# Genotoxicity Evaluation of The Novel Psychoactive Substance MTTA

**DOI:** 10.3390/ijms241310498

**Published:** 2023-06-22

**Authors:** Monia Lenzi, Sofia Gasperini, Giorgia Corli, Matteo Marti, Patrizia Hrelia

**Affiliations:** 1Department of Pharmacy and Biotechnology, Alma Mater Studiorum University of Bologna, 40126 Bologna, Italy; sofia.gasperini4@unibo.it (S.G.); patrizia.hrelia@unibo.it (P.H.); 2Department of Translational Medicine, Section of Legal Medicine, LTTA Center and University Center of Gender Medicine, University of Ferrara, 44121 Ferrara, Italy; giorgia.corli@unife.it (G.C.); matteo.marti@unife.it (M.M.); 3Collaborative Center for the Italian National Early Warning System, Department of Anti-Drug Policies, Presidency of the Council of Ministers, 00186 Rome, Italy

**Keywords:** MTTA, mephtetramine, cathinones, novel psychoactive substances, stimulant, genotoxicity, in vitro mammalian cell micronucleus test, flow cytometry

## Abstract

MTTA, also known as mephtetramine, is a stimulant novel psychoactive substance characterized by a simil-cathinonic structure. To date, little has been studied on its pharmaco-toxicological profile, and its genotoxic potential has never been assessed. In order to fill this gap, the aim of the present work was to evaluate its genotoxicity on TK6 cells in terms of its ability to induce structural and numerical chromosomal aberrations by means of a cytofluorimetric protocol of the “In Vitro Mammalian Cell Micronucleus (MN) test”. To consider the in vitro effects of both the parental compound and the related metabolites, TK6 cells were treated with MTTA in the absence or presence of an exogenous metabolic activation system (S9 mix) for a short-term time (3 h) followed by a recovery period (23 h). No statistically significant increase in the MNi frequency was detected. Specifically, in the presence of S9 mix, only a slight increasing trend was observable at all tested concentrations, whereas, without S9 mix, at 75 µM, almost a doubling of the negative control was reached. For the purposes of comprehensive evaluation, a long-term treatment (26 h) was also included. In this case, a statistically significant enhancement in the MNi frequency was observed at 50 µM.

## 1. Introduction

Novel psychoactive substances (NPSs) have steadily increased in type and number over the last few years, establishing themselves as a social threat of worldwide concern [1]. These are compounds commonly marketed and used as “legal” replacements of common drugs of abuse (i.e., cocaine, amphetamine, MDMA, cannabis and heroin) [2].

In line with recent reports showing cocaine as one of the most abused illicit drugs, psychostimulants have gained high acclaim among the users’ population, thus representing one of the widest categories of NPSs [1]. In fact, large-scale confiscations of synthetic cathinones (SCs) have dominated seizures in Europe in the last few years [1] and have revealed the increased quality of illicit products containing these substances [3].

Although NPS production and export have been affected by the legal restrictions imposed to limit their spread, data have shown its rapid adaptation to these changes based on the peculiar dynamic nature of the market [1]. Further confirming this dynamism, studies have shown the rapid appearance and disappearance of several such substances [4]. Among these compounds of brief existence, 2-((methylamino)methyl)-3,4-dihydronaphthalen-1(2H)-one (mephtetramine, or MTTA) was first identified in 2013 in the United Kingdom [5], and was also detected in material seized in Italy [6] in the same year, before rapidly disappearing from the illicit drugs’ scenario [4]. MTTA has appeared on the market after the stoppage of the trade of synthetic analogues of cathinone imposed by the law based on the “analogues criterion” [4], thus possibly suggesting the aim of constituting a new generation of psychostimulants underlying its production [7]. One interesting research, published in 2021, reported MTTA as among the ten most abundant non-polymeric compounds with a maximum peak area identified in San Leandro Bay water samples [8].

In fact, MTTA is not strictly an SC, but a γ-aminoketone chemically related to SCs, and differs from them due to a further carbon between the amine and ketone group (Figure 1) [9].

Given its molecular scaffold, this compound has also been associated with conformationally rigid analogues of the antidepressant fluoxetine [7] that have shown appetite suppressant properties [10], also in line with effects induced by other psychostimulant NPSs [11]. Moreover, conflicting effects of MTTA have emerged from anecdotal descriptions on users’ Internet fora [12,13,14]. In fact, total ineffectiveness [12] and slightly increased talkativeness [13], as well as euphoria and mental stimulation up to hallucinatory state [14], have been linked to MTTA intake.

To date, the pharmaco-toxicological profile of MTTA has been scarcely investigated by the scientific literature and most MTTA studies concern its metabolism. In vitro and in vivo experiments have shown that this molecule undergoes multiple metabolic pathways whose products, as well as unmodified MTTA, were found in the blood, urine and hair of mice after MTTA administration [7,15]. One recent in vivo study considered the pharmaco-toxicological effects induced by MTTA following repeated administrations in mice, possibly confirming the mild stimulant profile of this atypical cathinone. Sensorial inhibition, a transient decrease in breath rate and body temperature and both a dose-dependent increase and biphasic effect on locomotor activity were pointed out for the first time; furthermore, important changes in the blood cells count and blood and urine physicochemical profile, as well as histological changes in heart, kidney and liver samples, were revealed in blood and urine samples from the same animals [16].

These findings emphasize the urgent need to further investigate the pharmaco-toxicological profile of this compound, with particular reference to its potential cytotoxic and genotoxic effects yet to be examined. Moreover, the genotoxicity of some SCs (3-MMC, 4-MEC, mephedrone, mexedrone, α-PVP and α-PHP) has already been tested and demonstrated [17,18,19], possibly suggesting such an effect also for the structurally related compound MTTA.

For this purpose, human lymphoblastoid TK6 cells were treated with increasing concentrations of MTTA from 0 to 100 µM and the cytotoxicity induced was analyzed in terms of viability, cellular replication and apoptosis. Afterwards, based on the results obtained, we selected concentrations suitable for the genotoxicity evaluation, analyzed in terms of the ability to induce structural and numerical chromosomal aberrations, by means of a cytofluorimetric protocol, developed in our laboratory, of the “In Vitro Mammalian Cell Micronucleus (MN) test”.

## 2. Results

### 2.1. Short-Term Treatment

In the early phases of this research, cytotoxicity and cytostasis tests were performed in order to select the most suitable MTTA concentrations for the subsequent genotoxicity experiments, i.e., the “In Vitro Mammalian Cell Micronucleus Test”, OECD guideline n°487 [20]. According to OECD guideline n°487, to assess the genotoxic potential of a substance through the micronuclei (MNi) frequency evaluation, the highest tested concentration should not induce a cytotoxicity greater than 55 ± 5% compared to that observed in the concurrent negative control; consequently, cell viability and cell proliferation rates in treated cells should be at least 45 ± 5% [20].

TK6 cells were treated with MTTA in the concentration range 0–100 µM for a short treatment time (3 h), with or without an exogenous metabolic activation (S9 mix). After treatment, TK6 cells were cultured in a fresh medium for a recovery period of 23 h to allow TK6 cells to complete a total of 1.5–2.0 cell cycles.

Bar charts reported in Figure 2 display cell viability after MTTA treatments. The recorded viability rates proved that all treatment conditions were far above the OECD threshold (represented by the red lines) and therefore suitable for the genotoxicity test.

The correct selection of the concentrations also requires us to monitor cytostasis; indeed, it is necessary that a sufficient number of cells are able to proliferate and transmit the genetic damage to the daughter cells. Thus, the OECD guideline suggests calculating the relative population doubling (RPD) to estimate proliferation in each experimental condition and fixes the threshold at 45 ± 5% [20]. As reported in Table 1, in both conditions, all concentrations respected the OECD threshold.

Starting from cytotoxicity and cytostasis results, 0, 50, 75 and 100 µM MTTA concentrations were selected for the short-term treatment and a further analysis of the apoptosis induction was carried out on them.

Indeed, to further deepen the cytotoxicity evaluation, we considered another important mechanism of a cell death alternative to necrosis, which could also generate false positives. In fact, apoptotic bodies could be “mistaken” and erroneously counted as MNi by the instrument. Therefore, it is advisable to evaluate the genotoxicity at concentrations that do not induce a doubling in the percentage of apoptotic cells compared to the negative control [21]. As shown in Figure 3, the induction of apoptosis doubled only at the greatest concentration (100 µM) in the absence of S9 mix. Therefore, only this condition was excluded for the MN test.

Thereby, the concentrations selected to be tested for the MNi frequency evaluation were 50 and 75 µM in the absence of S9 mix and 50, 75 and 100 µM in the presence of S9 mix. Cultures treated with MTTA were compared to untreated negative controls and to positive controls mytomicin C (MMC) and vinblastine (VINB), a known clastogen and aneuploidogenic agent, respectively; cyclophosphamide (CP) and benzo[a]pyrene (BaP), known clastogens that require metabolic activation (Figure 4).

The OECD guideline states that a test chemical is considered to be clearly positive if, in any of the experimental conditions examined, at least one of the test concentrations exhibits a statistically significant increase compared to the concurrent negative control [20]. Our results show no statistically significant increase in the MNi frequency at any condition. However, without S9 mix, at 75 µM, almost a doubling of the negative control was reached (Figure 4A–C) and, in the presence of S9 mix, only a slight increasing tendency was observable at all tested concentrations (Figure 4D). 

### 2.2. Long-Term Treatment

The OECD guideline states that, for a thorough evaluation, which allows us to conclude a negative outcome, a long-term treatment without metabolic activation should also be conducted (in addition to a short-term treatment with and without metabolic activation).

Similarly to the short-term treatment, cell viability analysis was performed on TK6 cells treated with different MTTA concentrations (25, 35, 50, 75 and 100 µM) for an extended treatment time (26 h). According to the OECD guideline threshold, the highest suitable MTTA concentration to be tested was 75 µM (Figure 5).

In addition, the RPD was calculated for each experimental condition after long-term treatments and the OECD threshold was respected up to 75 µM (Table 2).

Based on cytotoxicity and cytostasis outcomes, the concentration of 100 µM was discarded for the subsequent analyses of apoptosis.

As displayed in Figure 6, the induction of apoptosis never reached a twofold increase at any treatment condition compared to the negative control except for 75 µM MTTA-treated cultures, in which the increase doubled. For this reason, the concentration of 75 µM was excluded for the subsequent genotoxicity analysis.

MTTA concentrations of 35 and 50 µM were tested in the genotoxic analysis and compared to the untreated negative control and positive controls MMC and VINB (see Figure 7).

In contrast with short-term treatments results, after the long-term treatment, a statistically significant increase was observed at the highest concentration tested (50 µM, *p* = 0.0254), where a doubling in the MNi frequency was measured (Figure 7A). This increase can also be visualized in the representative dot plots (Figure 7B,C).

## 3. Discussion

To date, the information available on the genotoxic properties of NPSs is limited, as well as their potential long-term effects. With regard to the class of SCs, bibliographic searches conducted on the main databases (e.g., PUBMED) reported few publications regarding molecules of this class or extracts of the plant *Catha edulis*, which naturally contains the cathinone [17,18,19,22,23]. In particular, only three studies are available on the impact of SCs on DNA, including ours, in which we demonstrated how mexedrone is genotoxic already as a parental compound whereas two other cathinones, α-PVP and α-PHP, are genotoxic only after metabolic activation [19]. An in vivo study by Kamińska et al. showed that mephedrone caused oxidative DNA damage in the cerebral cortex of adult rats, assessed by an alkaline comet assay [18]. Lastly, regarding the molecules 3-MMC and 4-MEC, the genotoxic capacity was not proven in the Ames test for gene mutations in the absence or presence of S9 mix, while a positivity was obtained for the induction of DNA single and double-strand breaks through “single cell gel electrophoresis” (SCGE) in human buccal mucosa cells (TR146), but only in the absence of S9. In addition, 3-MMC resulted in a statistically significant induction of MNi. These two molecules did not cause oxidative damage to DNA through specific enzymatic tests (FPG and Endo III), however [17].

Regarding the molecule object of study, MTTA, to our knowledge, there is no information available about its genotoxicity. Therefore, the aim of our study was to fill this gap, in consideration of the possible serious long-term consequences on health.

The in vitro genotoxicity strategy reported by most guidelines includes a battery of tests able to detect the different types of genetic damage in order to completely characterize the genotoxic hazard. In this study, as a first step, the induction of chromosomal aberrations possibly induced by MTTA was evaluated by measuring the increase in the frequency of MNi. In fact, MNi are biomarkers of genetic damage and, in particular, of structural and numerical chromosomal aberrations.

MTTA-induced genotoxic potential was evaluated in accordance with OECD guideline n°487, which corresponds to the in vitro MN test [20]. To carry out a thorough investigation, this guideline recommends testing the genotoxic effects of not only the parental compound but also the corresponding metabolites produced in vitro in case the cells employed are not metabolically competent with respect to the test substances [20]. As assay system we selected human lymphoblastoid TK6 cells, a cell line validated for the MN test that has the advantage of being of human and non-tumoral origin, easy to culture and fast replicating. 

A crucial aspect of the cell line chosen for this study concerns the fact that TK6 cells do not express significative levels of metabolizing enzymes, which instead are important for the activation or detoxification of xenobiotics and therefore require an exogenous source of metabolizing enzymes.

In view of these aspects, TK6 cells were treated with MTTA at different concentrations (0, 50, 75, 100 μM) in the absence or presence of an exogenous source of metabolic activation, S9 mix, for a short-term treatment of 3 h. The treatment was then followed by a recovery time of 23 h in a fresh medium, which is the time needed to allow the cells to complete 1.5–2 cell cycles, necessary to fix and transmit the damage possibly suffered to the daughter cells.

After checking cytotoxicity, cytostasis and apoptosis induction to select the suitable concentrations, we treated TK6 cells and evaluated the MNi frequency.

The MN test was carried out according to an automated protocol published by our group by flow cytometry [21]. This platform offers numerous advantages compared to the classical method in microscopy since it guarantees a much more objective result that is not affected by the subjectivity of the interpretation of the operator, and it also analyzes a number of events ten times higher, allowing for a robust statistical analysis and therefore the detection of small increases in the MNi frequency. Moreover, it permits a considerable reduction in the analysis times.

Following the short-term treatment, this assay did not point out a statistically significant increase in the MNi frequency in MTTA-treated cultures compared to untreated controls under any experimental condition. The OECD guideline states that, for a thorough evaluation, which allows us to conclude a negative outcome, a long-term treatment without metabolic activation should also be conducted [20].

This result supports the need to continue the studies in order to verify whether the genotoxic effect appears or not after a long-term treatment.

For this reason, we performed the MN test by treating TK6 cells for 26 h with MTTA concentrations selected according to the same rationale previously described for the short-term treatments.

This prolonged exposure of TK6 cells to MTTA brought the MNi frequency fold increase to a statistically significant value at the highest concentration tested, i.e., 50 μM, allowing us to demonstrate the genotoxicity of MTTA and, in particular, its ability to cause chromosomal aberrations. For a complete evaluation of the genotoxic hazard, according to the tests battery strategy previously mentioned, it could be advisable to also carry out an assay for genic mutations (e.g., a bacterial reverse mutation test). However, this is especially required to confirm a negative genotoxic outcome, whereas it is certainly interesting but not necessary in the case of an already positive response for other validated genotoxicity endpoints.

Another reasonable question could be if the concentrations tested in this study are physiologically relevant and, in particular, if the one that has proven genotoxic is comparable to the ones detected in human samples after MTTA consumption. However, it is known that, for genotoxic substances, any dose is potentially toxic; therefore, it is not possible to define a no-observed-adverse-effect Level (NOAEL), i.e., a non-genotoxic dose. By decreasing the dose and/or limiting the exposure, the probability that DNA damage occurs simply decreases [24]. On the other hand, MTTA is an NPS, so it is currently difficult to hypothesize the exact doses assumed by consumers and, starting from an in vitro result, to predict the pharmacokinetic profile in humans.

An additional limitation of our study is that it does not allow us to define the mechanism underlying the genotoxic effect demonstrated. A hypothesis could regard MTTA’s chemical structure, although it could not be immediately considered among the so-called “alarm structures”. It could be noted that the MTTA molecule is quite non-polar and that the bicycle moiety has almost a planar tridimensional conformation. These features might suggest that MTTA can interact and damage DNA by intercalative insertion among base pairs. In fact, intercalation is a non-covalent phenomenon, mainly driven by hydrophobic interactions and favored by a suitable shape and size such as the presence of a large planar system and the absence of steric hindrance [25].

One final consideration regarding the significance of the S9 mix effect is also due. Indeed, in our study, only in the absence of S9 mix was a statistically significant cytotoxic, cytostatic and apoptotic effect recorded, but not in its presence. This demonstrates an effect consequent to the metabolic exogenous activation toward the MTTA molecule, but it is hasty to hypothesize its significance in vivo when starting from this result obtained in vitro. Indeed, it is worth remembering both sides of the coin: human metabolism could result in a detoxifying effect, but it could even be responsible for the production of additional genotoxic metabolites.

All in all, the ability of MTTA to damage DNA, considered along with its likely low cytotoxicity and the scarce ability to induce apoptosis, has important consequences and is alarming since the mutations suffered can be transmitted to daughter cells, possibly giving rise to several long-term consequences. This should remind us of the importance of testing all the different toxicity aspects for every substance for the purposes of an effective campaign aimed at raising awareness and alarm against old and new drugs of abuse.

## 4. Materials and Methods

### 4.1. Reagents

7-aminoactinomycin (7-AAD), annexin V-phycoerythrin (Annexin V-PE), benzo[a]pyrene (BaP), cyclophosphamide (CP), 20-70-dichlorodihydrofluorescin diacetate (DCFH-DA), dimethyl sulfoxide (DMSO), ethylenediaminetetraacetic acid (EDTA), fetal bovine serum (FBS), hydrogen peroxide (H_2_O_2_), L-glutamine (L-GLU), mitomycin C (MMC), Nonidet, penicillin–streptomycin solution (PS), phosphate-buffered saline (PBS), potassium chloride, potassium dihydrogen phosphate, propidium iodide (PI), Roswell Park Memorial Institute (RPMI) 1640 medium, BPC-grade water, sodium chloride, sodium hydrogen phosphate, vinblastine (VINB) (all purchased from Merck, Darmstadt, Germany), RNase A, SYTOX Green (purchased from Thermo Fisher Scientific, Waltham, MA, USA) and Mutazyme 10% S9 mix (purchased from Moltox, Boone, NC, USA) were used.

### 4.2. MTTA

MTTA was purchased from LGC Standards (LGC Standards S.r.L., Sesto San Giovanni, Milan, Italy). The substance was dissolved in absolute ethanol at a maximum concentration of 10 mM and stored at −20 °C. In order to avoid potential solvent toxicity, the concentration of absolute ethanol was kept in the range of 0.25–1% for all the experimental conditions.

### 4.3. Cell Culture

All experiments were carried out on human lymphoblastoid TK6 cells. Among the possible OECD-validated cells suitable for analyzing the presence of MNi, this cell line was selected because of its human and non-tumoral origin, ease of maintenance in culture and replicative speed [20,26]. 

TK6 cells were purchased from ATCC (Manassas, VA, USA) and were grown at 37 °C and 5% CO_2_ in complete medium consisting of RPMI-1640 supplemented with 10% FBS, 1% L-GLU and 1% PS. Considering TK6 specific time to complete a cell cycle (13 h), the exponential growth of the cell culture was maintained by diluting cells every two days in fresh medium. Cell density never exceeded the critical value of 9 × 10^5^ cells/mL.

### 4.4. Test Conditions

For all the analyses, TK6 cells were treated for 3 h followed by a 23 h recovery in fresh medium or for 26 h. The total of 26 h corresponds to approximately 1.5–2.0 normal cell-cycle lengths for TK6 cells, as suggested by OECD guideline n°487 [20].

Moreover, to consider the in vitro effects of both the parental compound and the related metabolites, TK6 cells were treated with MTTA in the absence or presence of an exogenous metabolic activation system, as indicated by OECD guideline n°487. Specifically, in this study, a cofactor-supplemented post-mitochondrial fraction (S9 mix) was employed, which contains a liver enzymatic cocktail derived from rats treated with enzyme-inducing agents, such as Aroclor 1254.

The final concentration of S9 mix in the culture medium was 1%.

Since OECD guideline recommends limited cell exposure to S9 mix [20], MTTA genotoxic potential was evaluated after the short treatment time previously described (3 h ± S9 mix), followed by 23 h of recovery in fresh medium. However, a prolonged treatment of 26 h without metabolic activation was included to carry out a more comprehensive assessment [20].

#### 4.4.1. Selection of Concentrations

To identify the adequate concentrations to be tested for the MNi frequency evaluation, the OECD guideline n°487 defined a threshold for cytotoxicity and cytostasis equal to 55 ± 5% compared to the negative control. Consequently, viability and proliferation should be at least equal to 45 ± 5% [20].

#### 4.4.2. Measurement of Cytotoxicity

In order to assess cytotoxicity, aliquots of 2.5 × 10^5^ TK6 cells were seeded and treated with MTTA 0, 50, 75 and 100 μM for 3 h with and without S9 mix, followed by 23 h of recovery, or with MTTA 0, 25, 35, 50, 75 and 100 μM for 26 h without S9 mix. After treatments, cytotoxicity was assessed by considering the viability percentage. In more detail, cells were stained with PI dye. The viability was automatically calculated considering 1000 events (cells) by the Guava software guavaSoft™ 2.7 (Merck, Darmstadt, Germany). For each treatment concentration, the percentage of viability was recorded and normalized to the concurrent negative control set at 100%.

#### 4.4.3. Measurement of Cytostasis

Aliquots of 2.5 × 10^5^ TK6 cells were treated with MTTA 0, 25, 35, 50, 75 and 100 μM for 3 h ± S9 mix followed by 23 h recovery or with MTTA 0, 25, 35, 50, 75 and 100 μM for 26 h and cytostasis was checked as follows. The number of seeded cells and that present at the end of the treatment was automatically calculated by the Guava software guavaSoft™ 2.7 (Merck, Darmstadt, Germany) and it allowed us to check cell replication by calculating population doubling (PD) (Equation (1)):(1)$$\mathrm{PD}\;=\log\left(\;\frac{\mathrm{post}-\mathrm{treatment}\;\mathrm{cell}\;\mathrm{number}}{\mathrm{initial}\;\mathrm{cell}\;\mathrm{number}}\;\right)÷\log2$$

Thereafter, relative population doubling (RPD) was obtained by comparing the PD belonging to negative controls to that of treated cultures to verify that the majority of cells had completed cell division after the treatment (Equation (2)).
(2)$$\mathrm{RPD}\;=\frac{\mathrm{PD}\;\mathrm{in}\;\mathrm{treated}\;\mathrm{cultures}}{\mathrm{PD}\;\mathrm{in}\;\mathrm{control}\;\mathrm{cultures}}\times 100$$

#### 4.4.4. Measurement of Apoptosis

In order to identify the MTTA concentrations to be tested for the MNi frequency analysis, the apoptotic process was also evaluated as one of the cell death mechanism alternatives to necrosis. In particular, the concentrations tested were able to induce a two-fold increase in apoptosis compared to those registered in negative controls. Starting from the cytotoxicity and cytostasis analyses, aliquots of 2.5 × 10^5^ TK6 cells were treated with the selected MTTA concentrations 0, 50, 75 and 100 μM for the short-term time (3 h) and with MTTA 0, 25, 35, 50 and 75 μM for the long-term time (26 h).

The percentage of apoptotic cells was assessed by the double staining with two fluorophores: 7-AAD, which emits in red and is able to distinguish live cells from necrotic cells, and Annexin-V-PE, which allows us to quantify the apoptotic cells through a yellow emission. For this assay, 2000 events (cells) were considered.

The Guava software guavaSoft™ 2.7 (Merck, Darmstadt, Germany) automatically calculates the percentage of live, apoptotic and dead cells. The apoptotic cell percentage recorded in treated samples was normalized to that registered in negative control, whose value was set at 1 and expressed as apoptotic fold increase. 

#### 4.4.5. Measurement of MNi Frequency

Aliquots of 2.5 × 10^5^ TK6 cells were treated with different MTTA concentrations selected based on the results of cytotoxicity, cytostasis and apoptosis analyses. In more detail, 0, 50 and 75 μM MTTA concentrations were set for the short-term treatment in the absence of S9 mix, whereas 0, 50, 75 and 100 μM in the presence of S9 mix and 0, 35 and 50 μM for the long-term treatment.

MMC and VINB, known clastogen and aneuploidogenic agents, respectively, were used as positive controls when in the absence of S9 mix. On the contrary, the clastogens CP and BaP were used as positive controls that require metabolic activation when in the presence of S9 mix [20].

After treatment, the MNi frequency was evaluated by a flow cytometric protocol [21]. Briefly, aliquots of 7 × 10^5^ cells were collected, lysed and stained with SITOX Green. The MNi frequency was evaluated as the number of MNi per 10,000 nuclei derived from viable and proliferating cells. The discrimination between nuclei and MNi was based on their different size and different intensity of green fluorescence [21]. The MNi frequency recorded in treated cultures was then normalized to that recorded in the concurrent negative control cultures, set equal to 1, and expressed as MNi frequency fold increase.

#### 4.4.6. Flow Cytometry

FCM analyses were carried out with a Guava EasyCyte 5HT flow cytometer—class IIIb laser operating at 488 nm (Luminex Corporation, Austin, TX, USA).

#### 4.4.7. Statistical Analysis

All assays were repeated in three independent experiments and results are expressed as the mean ± SEM. Statistical significance was analyzed by paired analysis of variance for repeated measures (repeated measures ANOVA) or the mixed-effects analysis in case of missing values, followed by Dunnett or Bonferroni post-test using Prism Software 9.0. F, *p* and R^2^ values relative to repeated measures ANOVA analysis and F and *p* values for the mixed-effects statistical analysis were reported in each figure or table caption. All statistical analyses were carried out according to GraphPad Prism manual.

## Figures and Tables

**Figure 1 ijms-24-10498-f001:**
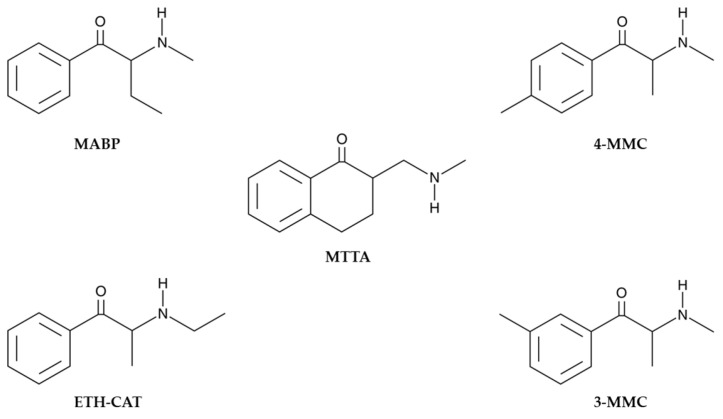
MTTA (γ-mephtetramine) and structurally related SCs: MABP (buphedrone), ETH-CAT (ethcathinone), 3-MMC (metaphedrone) and 4-MMC (mephedrone) (https://www.caymanchem.com, accessed on 12 June 2023).

**Figure 2 ijms-24-10498-f002:**
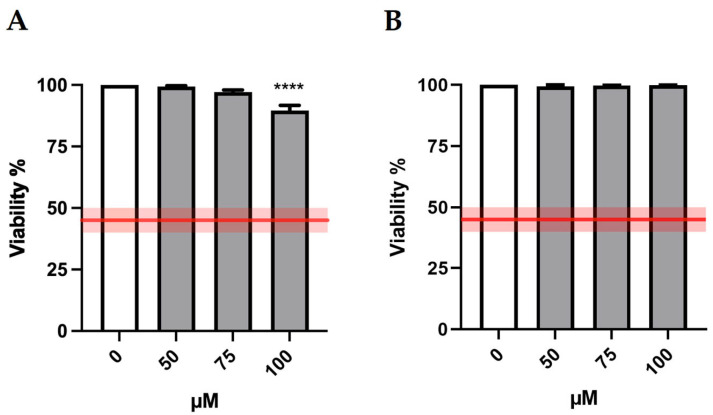
Cell viability of TK6 cells after 3 h treatment −S9 mix (**A**) or +S9 mix (**B**) at the indicated concentrations and compared to the concurrent negative control (0 µM). The red lines represent the OECD threshold for viability (45 ± 5%). At least three independent experiments were performed and the collected results are reported as the mean ± SEM. Data were analyzed by repeated measures ANOVA ((**A**): F (3, 15) = 20.27; *p* < 0.0001; R^2^ = 0.8022; (**B**): F (3, 15) = 0.8433; *p* = 0.4913; R^2^ = 0.1443) followed by Bonferroni post-test: **** *p* < 0.0001 vs. (0 µM).

**Figure 3 ijms-24-10498-f003:**
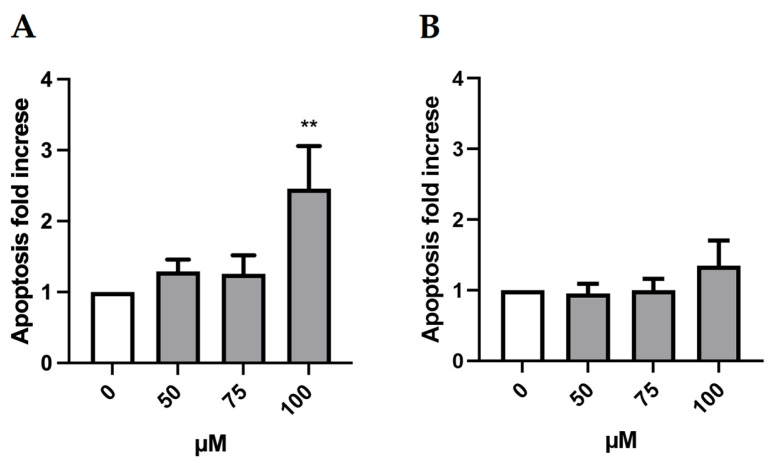
Apoptosis fold increase on TK6 cells after 3 h treatment −S9 mix (**A**) and +S9 mix (**B**) at the indicated concentrations and compared to the concurrent negative control (0 µM). At least three independent experiments were performed and the collected results are reported as the mean ± SEM. Data were analyzed by repeated measures ANOVA ((**A**): F (3, 15) = 5.614; *p* = 0.0088; R^2^ = 0.5289; (**B**): F (3, 15) = 0.7521; *p* = 0.5380; R^2^ = 0.1307) followed by Bonferroni post-test: ** *p* < 0.01 vs. (0 µM).

**Figure 4 ijms-24-10498-f004:**
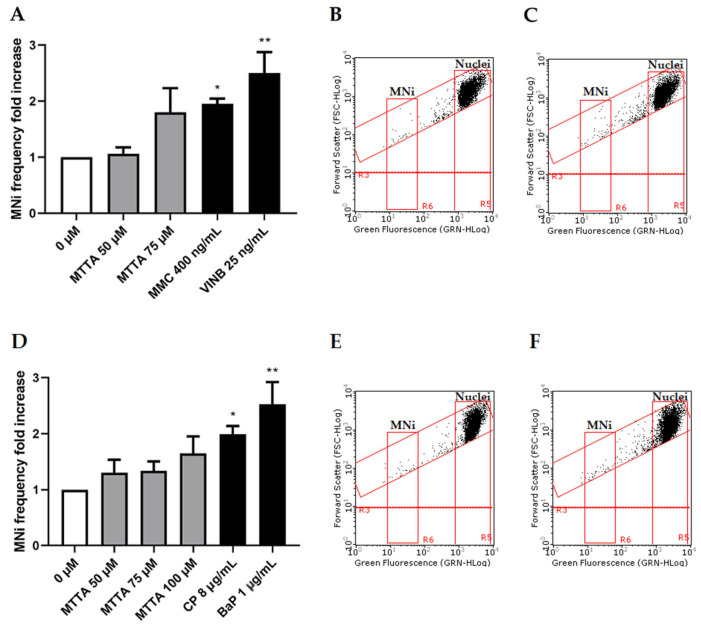
MNi frequency fold increase on TK6 cells after 3 h ± S9 mix treatment with MTTA followed by a 23 h recovery period. (**A**) Bar chart of the MNi frequency fold increase at different MTTA concentrations (50 and 75 µM) −S9 mix, compared to both the negative control (0 µM) and positive controls (MMC 400 ng/mL and VINB 25 ng/mL); (**B**) plot of nuclei and MNi in the negative control and (**C**) in 75 µM MTTA-treated cells −S9 mix; (**D**) bar chart of the MNi frequency fold increase at different MTTA concentrations (50, 75, 100 µM) +S9 mix, compared to both the negative control (0 µM) and positive controls (CP 8 µg/mL and BaP 1 µg/mL); (**E**) plot of nuclei and MNi in the negative control and (**F**) in 100 µM MTTA-treated cells +S9 mix. Each bar represents the mean ± SEM of three experimental replicates. Data were analyzed using repeated measures ANOVA ((**A**): F (4, 12) = 6.275; *p* = 0.0058; R^2^ = 0.6766; (**B**): F (5, 13) = 7.713; *p* = 0.0033; R^2^ = 0.7941) followed by Dunnett or Bonferroni post-test: * *p* < 0.05 vs. (0 µM); ** *p* < 0.01 vs. 0 (µM).

**Figure 5 ijms-24-10498-f005:**
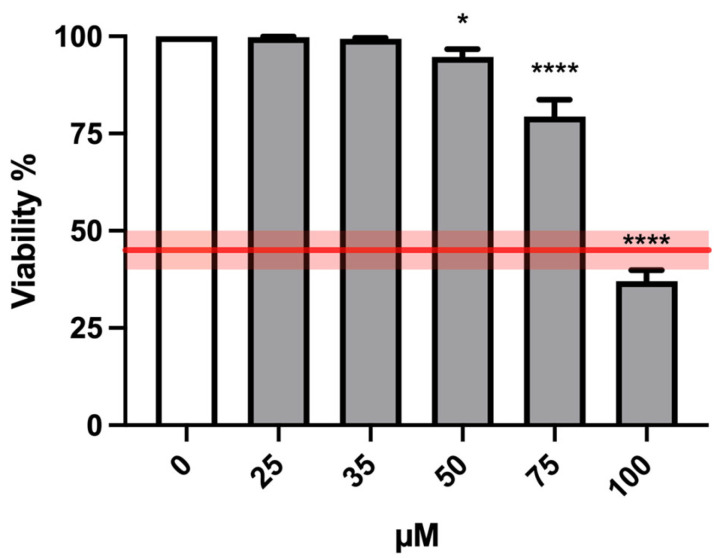
Cell viability of TK6 cells after 26 h treatment at the indicated concentrations and compared to the concurrent negative control (0 µM). The red line represents the OECD threshold for viability (45 ± 5%). At least three independent experiments were performed, and the collected results are reported as the mean ± SEM. Data were analyzed by mixed-effects analysis (F (5, 28) = 89.65; *p* < 0.0001) followed by Dunnett post-test: * *p* < 0.05 vs. (0 µM); **** *p* < 0.0001 vs. (0 µM).

**Figure 6 ijms-24-10498-f006:**
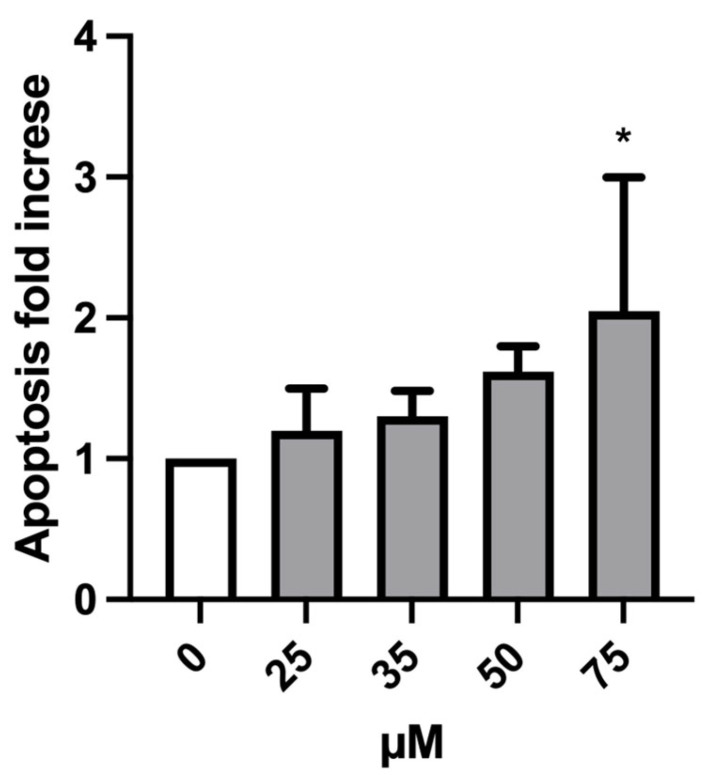
Apoptosis fold increase on TK6 cells after 26 h treatment at the indicated concentrations and compared to the concurrent negative control (0 µM). At least three independent experiments were performed and the collected results are reported as the mean ± SEM. Data were analyzed using mixed-effects analysis (F (4, 9) = 3.739; *p* = 0.0465) followed by Bonferroni post-tests: * *p* < 0.05 vs. (0 µM).

**Figure 7 ijms-24-10498-f007:**
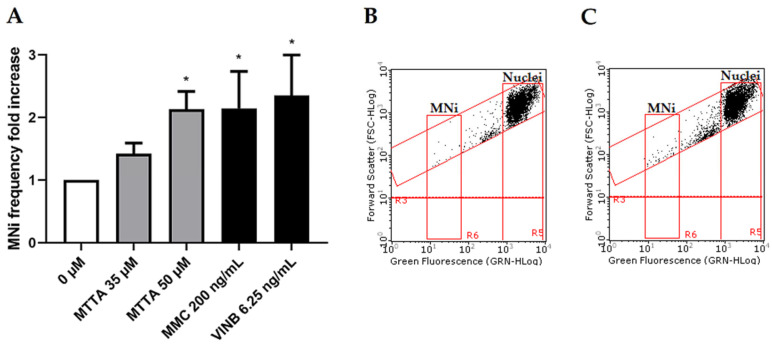
MNi frequency fold increase on TK6 cells after 26 h treatment with MTTA. (**A**) Bar chart for the MNi frequency fold increase at different MTTA concentrations (35, 50 and 75 µM) compared to both the negative control (0 µM) and positive controls (MMC 200 ng/mL and VINB 6.25 ng/mL); (**B**) plot of nuclei and MNi in the negative control and (**C**) in 50 µM MTTA-treated cells. Each bar represents the mean ± SEM of three experimental replicates. Data were analyzed using mixed-effects analysis ((**A**): F (4, 11) = 4.408; *p* = 0.0228) followed by Bonferroni post-test: * *p* < 0.05 vs. (0 µM).

**Table 1 ijms-24-10498-t001:** RPD on TK6 cells after 3 h treatment −S9 mix and +S9 mix with MTTA at the indicated concentrations compared to the concurrent negative control (0 µM). Each value represents the mean ± SEM of at least three independent experiments. Data were analyzed using repeated measures ANOVA (−S9: F (3, 15) = 33.98; *p* < 0.0001; R^2^ = 0.8717; +S9: F (3, 15) = 3.492; *p* = 0.0422; R^2^ = 0.4112), followed by Bonferroni post-tests: * *p* < 0.05 vs. (0 µM); ** *p* < 0.01 vs. (0 µM); **** *p* < 0.0001 vs. (0 µM).

Concentration	RPD 3 h −S9 (+23 h Recovery)	RPD 3 h +S9 (+23 h Recovery)
0 µM	100.0%	100.0%
50 µM	92.82 ± 2.29%	96.67 ± 1.60%
75 µM	79.22 ± 4.40% **	93.97 ± 2.08% *
100 µM	55.43 ± 7.32% ****	96.42 ± 1.71%

**Table 2 ijms-24-10498-t002:** RPD on TK6 cells after 26 h treatment with MTTA at the indicated concentrations compared to the concurrent negative control (0 µM). Each value represents the mean ± SEM of at least three independent experiments. Data were analyzed using mixed-effects analysis (F (5, 30) = 48.31; *p* < 0.0001) followed by Dunnett post-tests: *** *p* < 0.001 vs. (0 µM); **** *p* < 0.0001 vs. (0 µM).

Concentration	RPD 26 h
0 µM	100.0%
25 µM	91.03 ± 4.03%
35 µM	92.59 ± 2.27%
50 µM	81.21 ± 4.73% ***
75 µM	43.60 ± 9.32% ****
100 µM	0% ****

## Data Availability

The data presented in this study are available within the article.

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
