# Peer review of "Genotoxicity Evaluation of The Novel Psychoactive Substance MTTA"

_ijms, 2023, doi:10.3390/ijms241310498_

Round 1
Reviewer 1 Report
The article is innovative.
The abstract does not reflect the results obtained. It is necessary to indicate the most important results obtained.
For the analysis of variance, the effect size needs to be calculated.
It is worth saving the results of the analysis of variance according to scientific standards, e.g. F(2;34) = 5.63; p = 0.02; η2 = 0.14.
Limitations of the manuscript should be described in at least 10 sentences - there are quite a few.
It is worth dividing the discussion into logical subsections.
Minor editing of English language required.
Author Response
We thank the reviewer for the helpful suggestions to improve our work. The required information and changes are listed point by point and highlighted in red in the manuscript for faster viewing.
The article is innovative.
The abstract does not reflect the results obtained. It is necessary to indicate the most important results obtained.
We thank the reviewer for its observation. We corrected it.
For the analysis of variance, the effect size needs to be calculated.
It is worth saving the results of the analysis of variance according to scientific standards, e.g. F(2;34) = 5.63; p = 0.02; η2 = 0.14.
As suggested by the reviewer, we included in each figure and table caption the F and p values relative to Repeated Measures ANOVA or Mixed-effects statistical analyses. As regards η2, the software we use for statistics, i.e., GraphPad Prism, does not calculate it, but it shows the related value R2 for Repeated Measures ANOVA, which we reported.
Moreover, in the text we better explained that, as concerns cytotoxicity, cytostasis, apoptosis analyses, we had to check that the results respected or not the OECD thresholds to correctly select concentrations. Statistical significance, in our in vitro study, is only necessary to define a clear genotoxic response compared to the concurrent controls; therefore, we reported the p value for the the test chemical concentration that induced a statistically significant MNi frequency (lines 208-209).
Limitations of the manuscript should be described in at least 10 sentences - there are quite a few.
We added some other sentences in the Results and Discussion section to describe the limitations of the manuscript (lines 252-254, 283-288, 289-297, 298-299, 310-314).
It is worth dividing the discussion into logical subsections.
We thank the reviewer. According also to the other reviewer’s suggestions, we reduced and reorganized the whole discussion section.

Reviewer 2 Report
Lenzi et al assessed the genotoxic potential of a new psychoactive substance -MTTA- using the micronucleus test, which is a well-known biomarker of DNA damage and chromosomal instability. The topic is timely, relevant, and falls within the scope of the IJMS journal, yet the study design is remarkably similar to previous IJMS publications by the same authors on NPS (10.3390/ijms22126320; 10.3390/ijms21249616; 10.3390/ijms21031150).
-The authors allude to having "demonstrated...a weak increase in the MNi frequency" “…slight enhance has been recorded for the highest MTTA concentration…” in the abstract/results/discussion sections, however these findings are not statistically significant and must thus be rewritten (it is only a tendency) to avoid misleading the reader.
- Introduction section: please provide the chemical structures of mephtetramine and most related synthetic cathinones for comparison.
- Results section: Why was the concentration of 100 uM MTTA not chosen to be tested for MNi frequency if it was the only concentration to exhibit statistically significant increase in cytotoxicity (but far below the OECD threshold) and apoptosis on TK6 cells after a short-term (3h) treatment?
-The Discussion part must be considerably revised or combined into “results & discussion” because too many topics/explanations are repeated.
- Are the MTTA concentrations being evaluated physiologically relevant? Please bring this issue into discussion.
- Some discussion of the significance of metabolism (S9 mix) to the cytotoxic and genotoxic effects of MTTA is required.
Author Response
We thank the reviewer for the helpful suggestions to improve our work. The required information and changes are listed point by point and highlighted in red in the manuscript for faster viewing.
Lenzi et al assessed the genotoxic potential of a new psychoactive substance -MTTA- using the micronucleus test, which is a well-known biomarker of DNA damage and chromosomal instability. The topic is timely, relevant, and falls within the scope of the IJMS journal, yet the study design is remarkably similar to previous IJMS publications by the same authors on NPS (10.3390/ijms22126320; 10.3390/ijms21249616; 10.3390/ijms21031150).
-The authors allude to having "demonstrated...a weak increase in the MNi frequency" “…slight enhance has been recorded for the highest MTTA concentration…” in the abstract/results/discussion sections, however these findings are not statistically significant and must thus be rewritten (it is only a tendency) to avoid misleading the reader.
We thank the reviewer for the observation. We rewrote the results and modified the abstract and the discussion accordingly.
- Introduction section: please provide the chemical structures of mephtetramine and most related synthetic cathinones for comparison.
We included a figure (Figure 1) of the chemical structures of mephtetramine and most related synthetic cathinones in the introduction section.
- Results section: Why was the concentration of 100 uM MTTA not chosen to be tested for MNi frequency if it was the only concentration to exhibit statistically significant increase in cytotoxicity (but far below the OECD threshold) and apoptosis on TK6 cells after a short-term (3h) treatment?
We thank the reviewer for this observation, which gave us the opportunity to reformulate the rationale underlying the concentrations selection for the genotoxicity test more clearly (lines 105-108, 120-121 and 132-139).
-The Discussion part must be considerably revised or combined into “results & discussion” because too many topics/explanations are repeated.
We thank the reviewer. According also to the other reviewer’s suggestions, we reduced and reorganized the whole discussion section.
- Are the MTTA concentrations being evaluated physiologically relevant? Please bring this issue into discussion.
We thank the reviewer for this observation. We included a comment on this aspect in the discussion sections (lines 289-297).
- Some discussion of the significance of metabolism (S9 mix) to the cytotoxic and genotoxic effects of MTTA is required.
We thank the reviewer for this observation. We included a comment on this aspect in the discussion sections (lines 307-314).

Round 2
Reviewer 2 Report
The authors have addressed my concerns and made the required modifications, however the scientific soundness of the work, in my opinion, is not good enough for IJMS criteria. I will leave this decison to the Editor of this SI.